# Adolescent Social Media Use through a Self-Determination Theory Lens: A Systematic Scoping Review

**DOI:** 10.3390/ijerph21070862

**Published:** 2024-06-30

**Authors:** Monique West, Simon Rice, Dianne Vella-Brodrick

**Affiliations:** 1Centre for Wellbeing Science, University of Melbourne, Melbourne 3010, Australia; dianne.vella-brodrick@unimelb.edu.au; 2Centre for Youth Mental Health, University of Melbourne, Melbourne 3010, Australia; simon.rice@unimelb.edu.au

**Keywords:** social media, wellbeing, adolescence, self-determination theory

## Abstract

Background: Social media is an integral part of adolescents’ lives and has a strong influence on development and wellbeing. Research examining adolescent social media use and wellbeing is confusing as findings are inconsistent, inconclusive and contradictory. To address this issue, digital wellbeing scholars recommend that researchers adopt a theoretical approach with the aim of increasing meaningfulness and applicability of findings. Hence, this review applies self-determination theory to investigate how adolescent social media use supports and thwarts the basic psychological needs of relatedness, autonomy and competence. Satisfaction of all three psychological needs is essential for optimal development and wellbeing. Methods: A scoping review was conducted using a systematic search of five databases relating to adolescent social media use. The preferred items for systematic review and meta-analysis protocols (extension for scoping reviews) was applied resulting in 86 included studies. Results: Adolescent social media use both supports and thwarts relatedness, autonomy and competence. The findings highlighted how different aspects of adolescent social media use (including intra-personal, inter-personal, situational and environmental factors) contribute to the satisfaction and frustration of basic psychological needs. Conclusions: This review illustrates how social media can be both beneficial and detrimental to satisfying the basic psychological needs of relatedness, autonomy and competence. This is important when considering that if psychological needs are satisfied or frustrated in adolescence, the repercussions can have a cascading effect throughout adulthood. This review identifies gaps in the literature and provides suggestions for future research.

## 1. Introduction

The global uptake and pervasive integration of social media within adolescents’ everyday lives raises questions as to how social media impacts their wellbeing [1,2,3]. An extensive body of research has accumulated including reviews and meta-analyses (e.g., [3,4]). However, conflicting and contradictory findings make it difficult to draw conclusions from the literature [3,4,5]. The rapidly-evolving and dynamic nature of social media, coupled with the multifarious behaviours and experiences of social media users, creates challenges when endeavouring to compare results. Prominent digital wellbeing scholars have identified approaches to advance the field [3,6,7]. One recommendation is that researchers apply a theoretical framework to generate increasingly meaningful insights and applicability of findings [5,8]. A further suggestion is to adopt a developmental approach to capture the complexity and nuance of social media use at different life stages [3,4,6,7,8].

Consistent with these recommendations, the present review provides an overview of the current literature focusing specifically on *adolescent* social media use, and analyses findings through a self-determination theory (SDT) framework [9]. It includes a comprehensive examination of both positive and negative implications of adolescent social media use with regards to the basic psychological needs identified within SDT: relatedness, autonomy and competence. This systematic scoping review also highlights gaps in the literature and identifies focus areas that warrant attention within future research.

### 1.1. Social Media and Wellbeing

Wellbeing is a complex construct; consequently, within social media research, scholars often adopt different conceptualisations and measurements of wellbeing [10]. Some studies have focused on broad wellbeing definitions including happiness, psychological wellbeing or life satisfaction [11,12,13,14]. Others focus on specific indicators of wellbeing such as positive and negative affect, loneliness, self-esteem and social-connectedness [15,16,17,18]. Meta-analyses have aggregated wellbeing or illbeing outcomes [19,20,21] with some studies combining wellbeing and illbeing domains to determine an overall wellbeing score [10,22]. Such diverse approaches to defining and measuring wellbeing create ambiguity when endeavouring to interpret and draw conclusions from the literature, which may help explain ongoing controversies and confusion surrounding social media use implications for wellbeing.

Historically, research in this field has predominantly focused on the negative consequences of social media [1,5]. As such, there is a large body of work that adopts a deficit-based wellbeing approach. A common line of inquiry examines the relationship between social media use and mental health concerns including self-harm, suicidal ideation, negative body image, depression and anxiety [23,24,25,26,27]. Extensive literature explores a broad range of risk experiences such as exposure to harmful content (e.g., cyberbullying, hate speech, incivility or misinformation) [28,29,30], experiences of FOMO (fear of missing out) [31], and the detrimental effects of social media fatigue and digital stress [32,33]. Moreover, an increasingly growing field of research investigates pathological social media use including social media addiction and dependency [34,35]. This body of work has provided valuable findings; however, caution needs to be taken to “avoid over-pathologising everyday behaviour” [2] (p. 5).

It has been long recognised that wellbeing and illbeing are independent constructs that can have different correlates, causations and consequences [36]. The two domains can co-exist; thus, experiences of illbeing may coincide with experiences of wellbeing. For example, a person may experience considerable stress from their job yet also find their job rewarding as it fulfils a sense of purpose. Furthermore, it cannot be assumed that high scores of illbeing reflect low levels of wellbeing and vice versa. Consistent with this notion, scholars have recognised the need to adopt a balanced approach that considers both the positive and negative implications of social media use for wellbeing [1,37,38,39].

To combat the disproportionate focus on negative aspects of social media use, there has been a recent shift in scholarly attention towards the positive implications [1,37,40]. This is exemplified by the emergence of the new field of research—*Digital wellbeing*—which while in its infancy has rapidly increased over the past few years [41]. Rather than medicalising digital technology use, Vanden Abeele’s [39] concept of digital wellbeing acknowledges that digital media can promote wellbeing through facilitating hedonic and eudaimonic experiences. *Positive media psychology* is another new field that adopts a positive psychology paradigm to better understand how media technologies (including traditional and digital forms) can support wellbeing [42]. Scholars within this field stress that, due to the permanence and wide-spread adoption of social media, research should draw attention to how social media can be optimised to foster wellbeing and help people to flourish [42]. The recent development of the Digital Flourishing Scale further substantiates the growing interest in positive aspects of social media use. The Digital Flourishing Scale assesses users’ positive perceptions as to the benefits of digital communication, capturing both hedonic and eudaimonic experiences [37]. The Digital Flourishing Scale conceptualises digital flourishing through an SDT framework. Each subscale of the Digital Flourishing Scale corresponds with the basic psychological needs of relatedness, competence and autonomy.

### 1.2. Why Self-Determination Theory?

Ryan and Deci [43] note that facts without theoretical extension have little prescriptive value. Hence, this review applies an SDT framework with the aim of translating findings into practical and actionable agendas. SDT is a well-established, evidence-based theory that has been widely used within diverse fields of research including education, health care, parenting and organisational psychology [43]. SDT has been extensively verified as a theory of development, motivation and wellbeing that is universally applicable, with strong translational value [43]. A key tenet of SDT is that relatedness, autonomy and competence are innate basic psychological needs that are essential for optimal development and wellbeing [44]. *Relatedness* refers to having a sense of belonging and social connectedness, *autonomy* encapsulates the will or volition to drive actions, and *competence* reflects an individual’s capacity and ability to effectively achieve desired goals [9]. SDT also provides a comprehensive taxonomy of motives that explain regulatory styles and drivers that underlie human behaviour [44]. In recent years, scholars have recognised that SDT can be useful towards gaining meaningful insights into how social media use impacts wellbeing [1,8,37,42]. This is evidenced by the adoption of SDT as the underpinning framework for the Digital Flourishing Scale [37]. The value and utility of SDT is also reflected in a recent systematic review by Gudka et al. [1] that applied a positive psychology framework (including SDT) to investigate how social media can promote flourishing for people across a broad age range.

The scholarly interest in SDT is understandable when considering its credibility as a well-established and robust theory that is universal across cultures and domains [43]. However, there are further reasons why SDT is particularly suitable for examining social media use and wellbeing. Firstly, rather than focusing on individual indicators of wellbeing, there is a call for research to adopt a more holistic and multidimensional approach [1,2,8]. Thus, SDT provides a broad theoretical and empirical framework that allows the connection and convergence of heterogenous findings. Secondly, according to SDT, wellbeing is supported when the basic psychological needs of relatedness, autonomy and competence are satisfied, yet stifled when these needs are frustrated [44]. Therefore, consistent with recommendations for researchers to consider beneficial *and* detrimental aspects of social media use [1,17], SDT offers a framework to explore both the positive and negative implications for wellbeing. Thirdly, a fundamental tenet of SDT is that “our propensities toward autonomy competence and relatedness and the flourishing associated with them require specific social nourishments and supports” [43] (p. 105). Thus, SDT’s emphasis on the significant contribution social environments play towards wellbeing outcomes makes SDT particularly fitting when considering how social media has become a pervasive social context.

### 1.3. Adopting a Developmental Lens

The popularity of social media amongst adolescents has exponentially increased and is firmly embedded within youth culture worldwide [45]. Although cross-country differences exist with regards to young people’s social media adoption and use [46], the eager uptake of social media and the pervasiveness within adolescents’ lives appears universal. For example, in the USA, 54% of adolescents aged 13 to 17 report it would be hard to give up social media [47]. In China, approximately 183 million teenagers use instant messaging [48]. More than 90% of young people aged 16 to 24 use social networking sites in Germany, Denmark and Sweden [49]. Adolescents invest a considerable amount of time using social media. In the USA, 46% of adolescents aged 13 to 17 are reportedly online ”almost constantly” and 36% say they spend too much time on social media [47]. In the UK, approximately 95% of 15-year-old adolescents use social media before and after school [50] and a fifth of adolescents aged 13 to 15 report spending at least five hours per day on social media [51]. Similarly, Australians aged 14 to 17 spend more than two hours on social media per day [52].

Digital wellbeing scholars acknowledge that research examining social media use and wellbeing should differentiate adolescent and adult experiences [7,53]. Unlike many adults who have adopted social media later in life, today’s adolescents have typically joined social media from a young age and are considered “the most digitally connected generation” [54] (p. 143). Social media has been omnipresent and seamlessly integrated within many domains of adolescents’ lives. It has drastically transformed the way young people navigate their everyday activities and social worlds [45]. For example, there have been shifts in the way adolescents maintain and develop relationships, seek and share information, regulate their moods, and spend their leisure time [45,55,56]. A range of more nuanced consequences of growing up with social media have been identified. For example, adolescent social media use is associated with greater materialism [57], increased social comparison [58], and heightened civic engagement [59]. Furthermore, older generations’ experiences with social media differ from adolescents [53,60]. Compared with adults, adolescents engage with social media more frequently, are more prone to using multiple platforms, have different motives for use, and access and share different content [53,61,62]. Subsequently, young people are experiencing adolescence very differently compared with older generations [53]. Their online and offline worlds are intricately interconnected; the distinction between ‘real world’ and ‘online world’ has dissipated [53]. Thus, when considering how entrenched social media is within many aspects of adolescents’ lives, it has the potential to profoundly impact their development.

Applying a developmental lens is critical when endeavouring to understand the implications of adolescent social media use [7,45]. During adolescence, young people experience rapid and substantial changes across multiple domains including biological, cognitive, psychosocial and emotional [63]. These changes do not occur independently; they are interrelated and strongly impacted by socio-contextual factors [64]. Furthermore, adolescence is characterised by heightened neural plasticity and malleability with unique sensitivity to environmental influences [64]. This positions adolescence as a critical window of opportunity where experiences and exposures have the potential to impact the developmental trajectory [65]. The recent development of the Digital Flourishing Scale for Adolescents reinforces this view [66]. The developers stressed that an adolescent version of the Digital Flourishing Scale was necessary as developmental changes unique to adolescence are “expressed in adolescents’ differential uses of digital communication” [66] (p. 3).

Positive development in adolescence can have a cascading effect into future life stages; unfortunately, negative development can do the same [67]. Social media can be both deleterious and beneficial for key developmental tasks that occur during adolescence such as identify formation, individuation, the acquisition of social skills and expansion of social worlds [54,56,68]. Therefore, research focusing specifically on adolescent social media use could reveal valuable information to help guide adolescents towards using social media in ways that promote positive development.

In recent years, there has been a growing number of literature reviews investigating adolescent experiences of using social media and the potential impact on wellbeing [69,70,71]. For example, a scoping review by Schønning et al. [70] sought to explain the complex relationship between adolescent social media use, mental health and wellbeing. They found that most studies adopted a deficit-based wellbeing approach and used time-based measurements of social media use. The authors suggested that future research should consider positive aspects of social media use and apply qualitative methods to capture in-depth insights. A recent systematic review by Senekal et al. [71] investigated adolescent social media use with relation to psychosocial development. Findings were based on quantitative studies and highlighted both detrimental (e.g., cyberbullying) and protective implications (e.g., interpersonal support) for wellbeing. In line with Schønning et al.’s recommendation to use qualitative approaches, Popat et al. [69] conducted a qualitative narrative review of adolescent perceptions of how social media impacts wellbeing. Findings aligned with that of Senekal et al., demonstrating both positive and negative implications for wellbeing. These reviews contribute to the current knowledge in the field emphasising the complex relationship between adolescent social media use and wellbeing.

### 1.4. The Present Review

Despite the growing interest in adolescent social media use and wellbeing, the lack of reviews adopting a theoretical framework is notable. This current review addresses this gap and uses SDT as a structured framework to conceptually and empirically connect findings from the diverse and disparate literature. An SDT perspective is particularly suitable for exploring a social phenomenon and lends itself to adopting a balanced approach that investigates both the positive and negative implications of social media use. Applying SDT as a guiding framework also provides a transparent and conciliant format allowing for ease of replicability for future research [43]. Furthermore, consistent with recommendations to apply a developmental perspective when exploring the impact of social media [7,45,53], this review focuses on *adolescent* social media use. Given that social contexts play a central role in adolescent development, and that social media is intricately woven within adolescents’ lives, exploring the literature that specifically pertains to adolescents is crucial [7,53]. Moreover, as the social media landscape is constantly evolving and research in this field rapidly accumulates, on-going and up-to date reviews are needed.

Scholars recognise that scoping reviews are a more suitable alternative to systematic reviews when addressing certain research aims [72,73,74]. The general purpose of a scoping review is to identify and map literature on a broad research topic which in turn will generate foundational evidence for further research [72]. As such, scoping reviews are particularly suitable for examining emergent research from heterogenous fields [74]. Conversely, systematic reviews focus on specific research questions and assess risk of bias and quality of evidence with the aim of producing statements that guide clinical decision making [75]. Scholars note that applying a systematic approach when conducting a scoping review ensures the process is thorough, rigourous and replicable [72,73]. The aims of the current study align with the indications and purposes of scoping reviews [72]. Systematic scoping review guidelines outlined by Peters et al. [73] were applied throughout the review process. The current review had two key aims. The first aim was to provide an overall picture of the current state of evidence with regards to how adolescent social media use thwarts and supports the psychological needs of relatedness, autonomy and competence. The second aim was to identify gaps in the literature that can be addressed within future research to increase understanding on how adolescent social media use impacts the psychological needs identified within SDT.

## 2. Methods

### Literature Search Strategy

A systematic search of electronic databases including Scopus, PsycINFO, Communication and Mass Media Complete, SocINDEX and Cumulated Index to Nursing and Allied Health Literature was conducted in March 2023. Literature was identified using combinations of search terms that specifically relate to the basic psychological needs identified within SDT: relatedness, autonomy and competence. Adolescents use a suite of social media platforms and applications [76]. Hence, within this review, social media refers to social networking sites, content sharing, and social online gaming. To capture a wide range of platforms/applications and activities associated with adolescent social media use, each term for social media included individual platforms as well as broad terminology (e.g., social media and online social network). The specific social media platforms and applications applied within the search were guided by search terms used within previously published systematic reviews in peer-reviewed journals that examined adolescent social media use and mental health and wellbeing [25,77]. In addition, a specialist librarian with expertise in this field provided guidance with search term refinement and development. The inclusion of librarians in the literature search process can improve the quality of literature reviews and increase rigour and reproducibility [78]. Table 1 provides an example of the search terms and descriptors used.

This scoping review drew on the preferred items for systematic review and meta-analysis protocols extension for scoping reviews (PRISMA-ScR) [74]. Covidence software program was used for screening and extraction of relevant studies. As can be seen in Figure 1, the literature search identified a total of 4137 potentially relevant studies, excluding duplicates. A standardised data extraction template was devised, piloted and refined. Eligibility criteria are displayed in Table 2. The age criteria was based on the World Health Organisation [79] definition of adolescence (10–19 years). The first author independently conducted initial screening of abstracts and titles which resulted in a shortlist of 139 studies. She then inspected the full texts against the inclusion and exclusion criteria. Studies that were questionable with regards to whether they satisfied the criteria were discussed with the third author and discrepancies were resolved. In contrast to PRISMA systematic review protocols, the PRISMA-ScR checklist considers assessing risk of bias an optional step for scoping reviews [74]. Accordingly, the current review did not include quality assessment of individual studies. However, one of the inclusion criteria was that studies must be published in academic peer-reviewed journals (as seen in Table 2). Therefore, each study has undergone a rigourous vetting process with the aim of upholding academic standards expected within scholarly journals. Eighty-six studies were included in the review. The first author categorised and analysed each study in line with the three basic psychological needs identified within SDT.

## 3. Results and Discussion

### 3.1. Characteristics of Included Studies

Of the 86 studies, 4 applied mixed-methods designs (representing approximately 4% of the total number), 29 (34%) adopted qualitative methods and 53 (62%) used quantitative approaches (see Table 3). Findings suggest that there has been growing interest in this topic with an increase in publications since 2019 (see Table 3). Among the included studies, very few focused on specific developmental stages of adolescence. Approximately two percent of studies focused specifically on early adolescence, nine percent on mid-adolescence and none of the studies focused solely on late adolescence. Most of the studies aggregated results across adolescent developmental stages (e.g., findings were commonly based on a broad age range such as 11–18 or 15–19 years). Three studies did not provide sufficient detail to determine the specific stages of adolescence from which they based their findings (e.g., the sample age was described as teenagers). The studies within this review encompass a diverse range of geographic regions reflecting how adolescent social media use on wellbeing is a global concern (refer to Figure 2). However, much of the research was concentrated in a few countries.

The Appendix A presents a summary of the 86 articles within this review and outlines which of the psychological needs (relatedness, autonomy and competence) each article addresses. It should be noted that some studies address more than one psychological need. Findings demonstrated that 68 studies correspond with relatedness (79%), 37 with autonomy (43%) and 43 with competence (50%). Furthermore, findings showed that adolescent social media use may simultaneously thwart and support the same psychological need. Three studies [80,81] specifically used SDT as the guiding framework for assessing and interpreting data. Chiang and Lin [80] examined whether adolescents’ psychological needs were satisfied following online gaming. Yang et al. [81] explored how social media multitasking is associated with the fulfilment of each of the basic psychological needs. Finally, a study published by the current authors focused specifically on adolescents’ social media use with regards to the psychological need for relatedness [56]. Although the remaining studies did not directly apply SDT, the constructs they explored reflect the psychological needs as conceptualised within an SDT framework (e.g., a sense of belonging and connectedness reflects relatedness and volitional actions, control over aspects of daily life reflects autonomy, and feeling capable and accomplishing meaningful goals reflects competence).

### 3.2. Findings through an SDT Lens

The following section discusses the literature with reference to each of the basic psychological needs identified within SDT: relatedness, autonomy and competence.

#### 3.2.1. Adolescent Social Media Use and Relatedness

The literature suggested that social media can play a central role in both supporting and thwarting relatedness. A key finding was that social media may support relatedness by facilitating bonding and bridging social capital acquisition. Many of the studies (e.g., [19,56,81,82,83,84,85,86,87]) note that adolescent social media use can deepen relationships and improve friendship quality which reflects bonding social capital. The reviewed literature demonstrated that social media can promote open communication and self-disclosure [87,88], encourage a greater degree of closeness with others [17,56,84,89] and can provide a valued context for meeting romantic partners [90,91]. Moreover, Riley et al. [92] found a positive association between adolescent social media use and empathic concern and perspective taking, which may contribute to accumulation of bonding social capital.

The literature also demonstrated that adolescents often use social media to expand their social networks which reflects bridging social capital [93,94,95]. Findings showed that a common reason adolescents use social media is to develop new friendships with people outside their usual peer groups [19,56]. For example, social media can encourage dissimilar people to connect via shared interests [56,96] and it defies physical barriers facilitating interaction between people from different geographical regions [56,86]. Social media can also prompt initiation of offline friendships that may not be pursued otherwise [56,97].

In addition to acquiring social capital, the literature demonstrated that adolescents may use social media to experience a sense of belonging and acceptance [18,56,98,99,100,101,102,103,104,105,106]. Findings showed that social media can allow adolescents to create social connections via shared identities [95]. When interviewed, adolescents discussed how social media can enhance their sense of community with peers at school [107]. Shepherd and Lane [108] found that social integration within school was related to adolescents’ social media adoption. Social media was commonly associated with or identified as a tool that facilitates social engagement and promotes reciprocal support and caring [88,89,109,110]. The literature highlighted that adolescent social media use may combat social isolation and feelings of loneliness [93,111] and help adolescents avoid social exclusion [51,102]. Furthermore, findings revealed that adolescents consider social media useful for gaining popularity and social stature [100,109,112].

The reviewed literature reinforces how the line between online and offline worlds is increasingly blurred due to social media [53]. Adolescents consider interactions on social media to be an extension of real-world relationships [110]. Evidence illustrates how social media enhances relational ties [33,84,112,113,114,115], allowing adolescents to develop, maintain and strengthen offline relationships [85,113,114,115]. Social media is also considered a valued tool for resolving social issues that transpire offline [116].

However, this review demonstrates that adolescent social media use is complicated with regards to how the psychological need for relatedness is supported. Antheunis et al. [83] found that although adolescent social media use was associated with increased bridging and bonding social capital, adolescents who engaged with social media for more than 40 h per week did not report these benefits. The complex nature of adolescent social media use is further emphasised when considering findings by Lee et al. [94] who found that social media use can differ for adolescents from different countries. Their study showed that Korean adolescents use social media more for monitoring and acquiring bridging social capital, whereas Australian adolescents use social media more for group activities and bonding social capital.

Despite the potential social media offers to support relatedness, the literature suggests that social media use may also be linked with factors that are detrimental. Dredge and Chen [117] found that adolescents who were heavy social media users experienced increased negative social interactions. Similarly, Legkauskas and Steponavičiūtė-Kupčinskė [118] demonstrated that the more time adolescents spent using social media during school classes, rather than engaging with the set activities, the poorer their relationships were with peers. Stresses associated with social media use were shown to create problems for friendships [108,119,120]. For example, the unrealistic pressures and expectations that some adolescents place on their peers to be available on social media causes strain on friendships and diminishes friendship closeness [56,120]. Findings demonstrated that adolescent social media use is linked with relational aggression [121], peer conflict [108], and may encourage poor behaviours compared with face-to-face social interactions [56,103] and incite offline confrontation amongst peers [121]. Furthermore, some studies demonstrated that social media use may contribute to adolescents feeling socially disconnected [17,22,122]. For example, Timeo et al. [122] illustrated how receiving fewer likes poses threats to adolescents’ sense of belonging and elicits feelings of being ignored and excluded.

#### 3.2.2. Adolescent Social Media Use and Autonomy

The literature highlights how adolescent social media use can potentially satisfy and frustrate the need for autonomy [80,81]. Findings demonstrated that social media may support autonomy by helping adolescents to experience a sense of control, self-governance and personal agency [98,110,111,115,123]. It allows adolescents to regulate their public life on their own terms [108]. It also provides a vehicle through which adolescents can actively acquire and disseminate information that is important to them, which contributes to a sense of empowerment [56,123]. Social media offers considerable choice, including opportunities to engage with a broad range of people, activities and apps that are particularly appealing to adolescents’ interests and support developmental needs [17,98].

One common theme within the literature was the potential for social media to influence autonomy regarding identity development. The research revealed that social media fosters a sense of liberty by providing opportunities for self-expression and identity construction [17,84,98]. For example, adolescents can control their online image, exert their creativity and curate personal profiles or posts, which reflect a version of the self that they choose to portray [19,86,123,124]. Furthermore, social media offers the flexibility to refine and revise online self-presentations as adolescents explore and experiment with different identities [124].

However, the literature also emphasised how adolescent social media use can challenge autonomy. Adolescents develop a sense of reliance on social media within many aspects of daily life. This was evident with regards to relying on social media to maintain and nurture relationships [84] and feeling emotionally dependent on social media [125]. The strong pull of social media was illustrated further in studies that highlighted how adolescents often experience FOMO (fear of missing out) when unable to access their social media [56,100,126,127].

Findings also suggested that other peoples’ actions and expectations may threaten adolescents’ autonomy when using social media. Similar to offline social systems, normative pressure and power-play exists on social media [98,102]. Adolescents sometimes feel limited and constrained with regards to their choices and actions on social media [98,128]. They often modify and edit their online behaviours, posts and comments to avoid negative repercussions from others [51,98,105,119,123,124]. Moreover, norms and peer practices create a sense of obligation (through fear of negative consequences) to be readily available, to reply promptly to posts and to leave positive feedback for friends [51,119,124,129,130].

Findings demonstrate that autonomy may be stifled by the lack of agency adolescents experience due to other peoples’ actions on social media. For example, adolescents sometimes feel embarrassed or hurt by unfavourable images or comments that other people post [86,99]. They are also impacted by a range of controlling behaviours including intrusive monitoring, location checking, and demands for photos [90,130,131]. In addition, at times adolescents feel threatened by unsolicited contact from strangers that pose a risk to privacy [130] or exposed to sexting abuse, coercion, blackmail, revenge distribution and bullying [110,131].

#### 3.2.3. Adolescent Social Media Use and Competence

Consistent with findings pertaining to relatedness and autonomy, the literature demonstrated how social media may potentially support and thwart competence. One topic of interest was how adolescents’ social media use can impact competence with regards to academic performance. On the one hand, research suggests that social media may be beneficial for adolescents’ educational outcomes. For example, Kasperski and Blau [88] interviewed adolescents and found that using social media for learning facilitation (e.g., administrative purposes, disseminating resources and learning support) extends the learning experience beyond the boundaries of school and promotes peer teaching. Alloway et al. [132] found that adolescents who had used social media for over 12 months had significantly higher scores in working memory, verbal ability and spelling compared to those who had used it for less time. Further studies showed that information seeking, peer-to-peer knowledge sharing, and accessing and disseminating learning-related resources via social media may enhance learning, improve academic achievement and positively predict academic performance [12,109,111]. In addition, critical thinking skills may be improved through seeking news via social media for informational purposes [133]. These studies are cross-sectional; therefore, it is unclear whether social media use enhances cognitive capacities or whether adolescents with greater levels of cognitive skills happen to use social media. However, it has been suggested that the opportunity to practice these skills via social media may potentially lead to a training effect offering positive cognitive benefits [132]. Furthermore, learning support from teachers and peers via social media may be especially helpful for introverted students enabling them to overcome learning barriers [88].

Nonetheless, the literature also demonstrates that adolescent social media use has the potential to hinder learning. Numerous studies showed an association between social media use and lower academic performance [12,19,134,135,136]. For example, adolescents who use social media for more than two hours a day reported lower academic achievement [135]. More time spent using social media during class was linked with an increase in missed classes and lower grade point average [118]. Poor time management associated with social media use was shown to be linked with reduced academic performance [19]. Tanrikulu and Mouratidis [126] found that checking social media during class creates distraction and study interference which may lead to lower grades and poor study efforts. Social media also distracts from activities that contribute to competence such as homework and sleep [111,137]. For example, Evers et al. [138] found an association between disturbed sleep (due to social media use) and lower academic achievement for Taiwanese middle school students.

Beyond the academic realm, the literature also highlighted that social media use can influence competence with regards to adolescents’ sense of self. The research demonstrated that social media can foster positive self-perceptions [85,93,109,113,139]. Studies showed that adolescents’ social media interactions can provide a form of affirmation, inspiration and ego validation [17,127]. It can increase confidence and self-esteem thus cultivating a positive self-concept [85,109,113,139,140]. A few studies discussed how positive perceptions are often prompted by feedback from others (e.g., receiving likes, peer attention or positive comments) that elicit feelings of affirmation and a sense of being valued [113,128,141]. Social approval and social competence is particularly important during adolescence as it is a stage of development where friendships have heightened importance and a strong influence on self-construction [63]. The literature outlined a variety of ways social competence is fostered by social media use such as strengthening communication, increasing empathic concern and perspective taking, encouraging friendship initiation and facilitating interaction between diverse people [85,92,111,142].

In contrast, detrimental consequences for adolescents’ sense of self were also evidenced within the literature. Studies revealed that adolescent social media use is associated with decreased self-esteem and confidence [113,128]. A broad range of potential threats to positive self-concept were identified including peer comparison, receiving fewer likes, stereotyping, insults, judgement, distress and cyberbullying [17,91,122,130]. Adolescents who check their social media more often reported increased emotional difficulties, worry, nervousness and fear [100]. Studies suggested that the constant gaze by others and scrutiny on social media may threaten a sense of competence by increasing self-consciousness [86]. A further finding was that adolescents with lower self-esteem felt more inclined to edit their posts due to fear of negative peer evaluation [128]. However, Chua and Chang [128] found that older adolescents tended to refrain from heavy editing and immense self-presentation efforts. Furthermore, Tsitsika et al. [136] found that heavy social media use by older adolescents was associated with increased social competence, yet this was not the case for younger adolescents. These findings align with recommendations by Orben et al. [4] who highlight the need to address the different stages of adolescence when investigating the impact of social media use.

A few studies identified the potential social media has to influence adolescents’ digital competence [71,128,142,143]. Overall, the research suggests that digital competence can be enhanced by social media use [128,143]. For example, using social media allows adolescents to increase their information and communication technology skills and knowledge [142]. Frequent social media use enables adolescents to hone their content editing skills [128] and supports gaming competence [80]. Furthermore, through watching other people make mistakes on social media, adolescents gain useful knowledge regarding peer normative expectations when using social media [128]. Adolescents who use social media also demonstrate a high level of digital safety competence (e.g., protection within devices and potential physical and psychological health risks) [143]. Despite the benefits for digital competence, heavy social media use may displace time spent on other activities thus challenging competence in different areas. For example, Tsitsika et al. [131] found an association between social media use and reduced competence in sporting activities and hobbies.

### 3.3. Limitations, Implications and Future Research

When interpreting the current findings, it is important to consider how exclusion criteria and database selection can limit review findings. As scholarly interest in social media spans diverse fields of research (e.g., media studies, developmental science and computer studies), many databases were suitable for this review. Nonetheless, despite the potential for some fields to be under-represented, the chosen databases were particularly relevant to both social media and wellbeing research and offered a comprehensive body of literature. Another point to consider is that studies within this review were published in English within academic peer-reviewed journals. Thus, some potentially relevant articles within grey literature or written in different languages may have been overlooked. Another consideration is that screening and data charting were conducted by a single author. As such, there is potential for error that may have been avoided if multiple reviewers were employed. Furthermore, due to the vast array of social media applications and platforms available globally, it is not plausible to include every option within the search descriptors. Taking this into account, broad search terms were applied (including social media and social networking sites) to capture a range of platforms. However, inevitably some platforms would have been missed or under-represented within the current search. Furthermore, many of the studies included in the review were cross-sectional which needs to be considered when interpreting findings. There was a notable lack of longitudinal studies which highlights a gap in the literature that warrants future research attention. Another consideration is that most studies within the digital wellbeing realm are based on clinical populations or outcomes [1,45]. It should be acknowledged that this review excluded research with a clinical focus to capture results that represent typical social media use amongst the general adolescent population. Nonetheless, to take into account the complexity of the social media phenomenon, further reviews could examine studies that are associated with clinical populations and/or problematic social media use and the potential association with basic psychological needs. This is important when considering the growing evidence demonstrating how social media behaviours (e.g., pre-occupation with using social media, excessive time spent on devices, multitasking, distraction from important tasks) and psychological vulnerabilities (e.g., emotion dysregulation, attention impulsiveness, FOMO, and alexithymia) are associated with problematic social media use amongst young people [144,145,146,147]. These factors can play a key role in whether basic psychological needs are thwarted or supported.

One of the inclusion criteria for the current review was that articles were based on adolescents aged 10–19 which is consistent with the WHO definition of adolescence [79]. However, developmental scientists stress that adolescence does not represent a single development phase [63]. Young peoples’ experiences at different stages of adolescence can vary greatly and should be considered when conducting research. Most of the studies within the current review reported results based on a broad age range within adolescence. Nonetheless, a few studies did apply narrow age perimeters. Approximately two percent focused on early adolescence, and nine percent on mid-adolescence; however, no studies focused solely on late adolescence. Therefore, future research could examine the behaviours and effects of social media use during early, mid and late adolescence. Delineating developmental stages could yield more nuanced insights and capture important information in relation to the different experiences with social media and wellbeing outcomes across adolescence [148].

A further consideration is that the current review presented findings for each basic psychological need separately. This approach provided a clear structure for synthesising and interpreting a wide range of papers. It also allowed each psychological need to be analysed and discussed in depth. However, it should be noted that the implications of social media use on psychological needs are often interconnected [44]. For example, when adolescents reach out to peers on social media, they can simultaneously deepen friendships, enhance social skills and elicit a sense of control. Considering the inherent ‘social’ underpinnings of social media, it is not surprising that most studies within the review corresponded with the basic psychological need of relatedness (79%) whilst fewer studies aligned with autonomy (43%) and competence (50%). Given that all three psychological needs are important for wellbeing [116], further investigation into the implications of adolescent social media use for autonomy and competence is warranted.

The current findings draw attention to the complexity of adolescent social media use and the need for scholarly research to further unpack the multiple layers at play. Notably, most of the studies identified within this review used quantitative approaches (refer to Figure 2). However, for the field of adolescent digital wellbeing to move forward, scholars recommend that researchers apply a greater diversity of methods; they note that qualitative approaches would be especially useful to capture in-depth and nuanced information. Furthermore, there is a strong call for research that champions adolescent voice, as consulting directly with adolescents is critical in research that aims to understand a social phenomenon through an adolescent developmental perspective [6,7,71].

Findings highlighted that scholarly attention focusing on this topic has increased over the past few years (see Table 3). This is not surprising considering the exponential uptake of social media amongst adolescents globally [45] and the worldwide prevalence of youth mental health issues [149]. Given that SDT is recognised as a universal theory that is applicable and relevant across cultures and countries [44], scholars from around the globe could benefit from applying an SDT framework within their work. Adopting an SDT perspective could generate important information on different experiences for adolescents from varying countries and cultures with regards to how social media impacts the fulfilment of basic psychological needs.

## 4. Conclusions

This systematic scoping review provides unique contributions to the current literature exploring adolescent social media use and wellbeing. Consistent with recommendations by digital wellbeing scholars, this review specifically captured experiences during adolescence which is a critical stage where social contexts have a strong influence on wellbeing and development [7,45,53]. This is the first review to examine adolescent social media use and wellbeing through an SDT framework providing a structured format to make sense of the disparate and divergent existing literature. It illustrated how the application of a multi-dimensional wellbeing framework can generate meaningful insights that have relevance and practical applicability. Findings provided examples of how adolescent social media use both supports and thwarts the psychological needs of relatedness, autonomy and competence. Future research should build upon this foundational evidence by further investigating the interactional and nuanced aspects of adolescent social media use and their potential to support and thwart the basic psychological needs. Using qualitative methods, championing adolescent voice and delineating the different experiences of early, mid and late adolescents could help towards gaining deeper insights. As social media is constantly evolving, prevalent within adolescents’ lives, and has the potential to strongly influence development and wellbeing, a continued research agenda within the field is critical.

## Figures and Tables

**Figure 1 ijerph-21-00862-f001:**
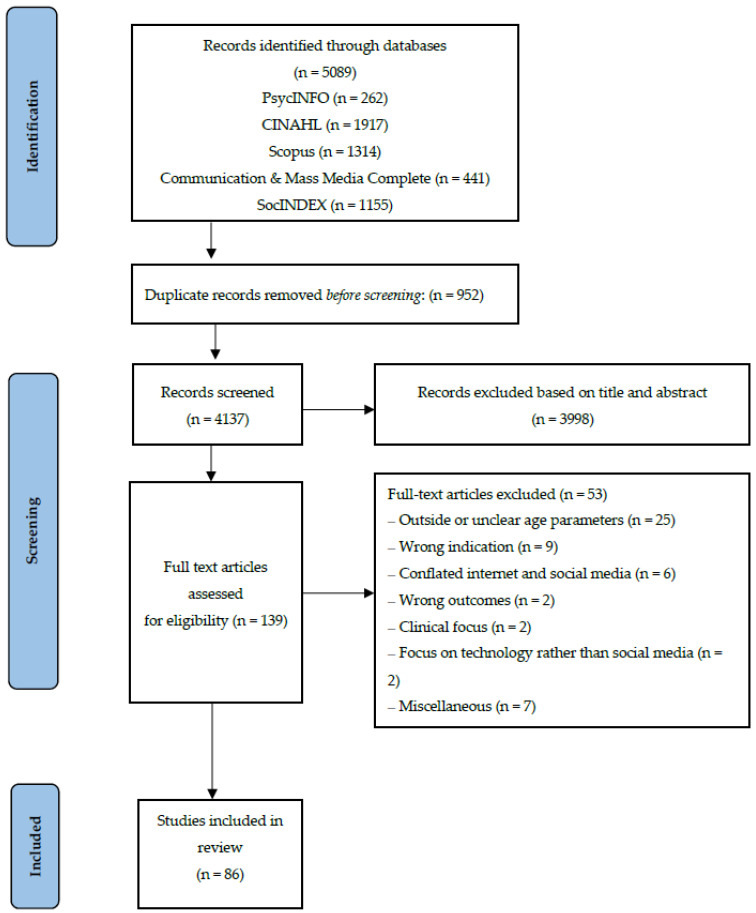
PRISMA flowchart illustrating literature search results.

**Figure 2 ijerph-21-00862-f002:**
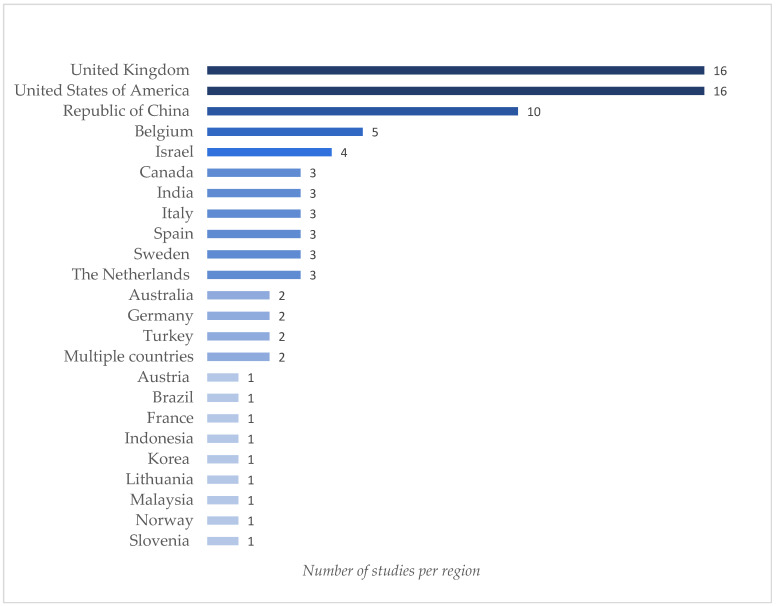
Geographic regions where studies were conducted and number of studies per region.

**Table 1 ijerph-21-00862-t001:** Literature search terms and descriptors.

Terms	Descriptors
#1 social media	(“social media” OR “social network*” OR “online social network” OR Facebook OR TikTok OR Instagram OR Insta OR “online gaming” OR Pinterest OR Snapchat OR “instant messag*”)
#2 self-determination theory	(“self-determination theory” OR self-determin* OR SDT OR “basic psychological need” OR relatedness OR relationship OR “social connect*” OR belong* OR “fit in” OR autonomy OR volition OR independen* OR “intrinsic motivat*” OR “extrinsic motivat*” OR “psychological empowerment” OR mastery OR self-efficacy OR efficacy OR competen* OR goal attainment OR “goal achievement” OR (need* N2 fulfil*) OR (need* N2 satisf*) OR (need* N2 frustrat*) OR (need* N2 thwart*) OR (need* N2 psychological))
#3 adolescents	(teen* OR tween* OR minor* OR adolescen* OR “young people” OR “young person” OR youth OR “generation z” OR “gen z” OR “digital native” OR “school student”)
combination	#1 AND #2 AND #3

**Table 2 ijerph-21-00862-t002:** Eligibility criteria.

Inclusion Criteria	Exclusion Criteria
Studies published in academic peer-reviewed journals with full text available in English Studies that focus on adolescents aged between 10 and 19 years Studies published between 2006 to current (it was not until 2006 that social media permeated popular culture)	Studies that focus on screen time or internet use rather than social media Studies based on clinical populations or outcomes Studies that aggregate findings across age groups beyond adolescence (10–19 years) Manuscripts that do not present research data based on original research

**Table 3 ijerph-21-00862-t003:** Publication year and study designs for the identified studies.

Publication Year	Studies per Year	Qualitative Studies	Quantitative Studies	Mixed-Method Studies
2007	1	-	1	-
2008	1	1	-	-
2009	1	-	1	-
2010	1	-	1	-
2011	1	-	1	-
2012	3	-	3	-
2013	4	-	4	-
2014	7	4	3	-
2015	6	2	3	1
2016	9	3	5	1
2017	7	2	5	-
2018	4	2	1	1
2019	10	4	5	1
2020	8	-	8	-
2021	12	6	6	-
2022	8	3	5	-
2023 (up to March)	3	2	1	-

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
