# Peer review of "Adolescent Social Media Use through a Self-Determination Theory Lens: A Systematic Scoping Review"

_ijerph, 2024, doi:10.3390/ijerph21070862_

Round 1

Reviewer 1 Report

Comments and Suggestions for Authors

This is a really interesting paper, in an important area of research: social media and mental health. I am particularly pleased that -as the authors’ note- it takes a balanced perspective on the risks and benefits of social media use, and approaches the topic through a theoretical lens, but also a theoretical lens that I think has real potential value to the field.

I think the introduction is extremely well written, and clear, and presents a really articulate and detailed overview of the relevant literature. Indeed, there are many strengths to the review, which I think should be published, as it will (eventually) make an important and valuable contribution to the field.

However, there are a number of issues I have with the paper as it currently stands, though I do not think these are insurmountable by any means. I include below some more specific points and details, but broadly, my main concerns are:

(i)                  The use of causal language or assumptions, that are not underpinned by (I suspect) much of the research reviewed.

(ii)                Missing information from the Methods, as well as the results (e.g., more detail as to the nature of papers included), without which it is not possible to critically evaluate the evidence base.

(iii)              A number of more minor methodological issues (described below).

Though I realise this is a scoping review rather than a systematic review, I think (i) in particular, as well as some of the other points noted below, risk muddying the water if they are not addressed.

Specific points:

·         Page 6: How were the platforms / apps included in the social media search terms selected / why were others not included? Were these selected on any principled basis, e.g., most common platforms used according to X? If not, this should be  justified and/or mentioned as a limitation, since it may have led to research on certain platforms / apps and research from other countries, e.g., China (including those published in English language journals), being missed.

·         The authors used the WHO definition of adolescence (10-19years), this should be made explicit in the Methods, however, not just in the discussion.

·         Because of a lack of conceptual clarity in the literature I think the authors should clarify their definition of “social media”, e.g.,: “Chiang and Lin (2010) examined whether adolescents’ psychological needs were satisfied following online gaming” (p.8-9).

·         P.10: “Their study showed that Korean adolescents use social media more for monitoring and acquiring bridging social capital, whereas Australian adolescents use social media more for group activities and bonding social capital.” Unless I am mistaken, the paper referenced (Lee & Hong, 2016) makes no reference to an Australian sample, as this sentence would seem to suggest. If the authors are just drawing comparisons across different papers (though this is not made explicit), this is I think inappropriate, since samples are likely to have differed on more than just country of origin.

·         More importantly however, Lee & Hong (2016) do not seem to make any reference as far as I can tell to “bridging & binding social capital” or even “social capital” for that matter, which is the reason that the paper was included according to the Table in the Appendix. Why was this paper included?

·         To aid the reader, the table in the Appendix should repeat the column headers on each page.

·         The papers seem to have been coded for a number of difference variables, e.g., age range included etc. but hardly any of this has been included in Tables (in the main body of the manuscript or in the Appendix). It would I think therefore be helpful to expand the table in the Appendix to include information about (for example), age of sample, gender balance of sample, country of origin / ethnicity of sample, date of data collection, etc. and for summary data from this to be reported in the main body of the paper also.

·         Did anyone second code / screen papers etc.? The process should be stated explicitly, e.g., was it only one person who did the data extraction phases etc.? If so this should also be noted as a limitation.

·         The review does not speak to study design (of included studies) at all and often falls into the trap of inferring / suggesting causality where the study design cannot make any such conclusions.

·         For example, the authors say that “despite the supportive aspects of social media use or relatedness, potential negative consequences were also identified within the literature.” (p.10). Thus, this sentence (and others throughout, e.g., “it contributes to a sense of community in school (Eklund & Roman, 2019; Shepherd & Lane, 2019)”) suggest a direction of causality running from social media use -> benefits and harms, whereas in fact, a reversed pattern is possible, e.g., people who feel more socially connected use social media in a different way, or people who feel more disconnected use social media in a different way, as some studies have shown.

·         As another example: “On the one hand, research shows that social media can be beneficial for adolescents’ educational outcomes. For example, Alloway et al., (2013) found that 452 adolescents who had used social media for over 12 months had significantly higher scores in working memory, verbal ability, and spelling compared to those who had used it for less time.” (I am not sure whether Alloway looked at correlations across time, but even these do not prove causation; instead, those with better working memory etc. may be more engaged / active and drawn to heavier social media use, for example).

·         Similarly, “critical thinking skills can be enhanced through seeking news via social media for informational purposes (Ku e. al., 2019)” cannot be inferred from cross-sectional data (which I think the referenced study draws upon).

·         I realise this is a scoping review, but I would still be tempted to: (i) code all the studies in terms of design (cross-sectional, longitudinal, experimental etc.), and (ii) reflect more critically on what the literature can / cannot say about underlying directions of causality as a result. At the very least I would recommend (ii).

·         Relatedly, I would include in your Supplementary table a description of the type of social media / platform / app that is the focus of each study, as well, potentially as the nature of the analyses undertaken (e.g., correlation, regression analyses etc.). Without this information it is hard to critique the papers / get a clear sense of them. For example, where the authors write “it contributes to a sense of community in school (Eklund & Roman, 2019; Shepherd & Lane, 2019)”, the former reference is actually to “digital gaming”, not social media per se, although as per my previous message, this is itself not clear since the authors do not provide a definition of social media in the paper.

·         The review includes papers that are not original research papers, e.g., Yang et al. (2021). This is problematic since the review may itself include papers that are included elsewhere in the review. I would therefore include as an exclusion criteria any papers that do not present research data, or at least highlight this.

Reviewer 2 Report

Comments and Suggestions for Authors

First of all, I thank and am honored to contribute to the revision of this work. I congratulate the authors for tackling an extremely important and topical issue, which certainly needs more attention.

Overall, I find one limit in this work that depend on dimensional and methodological choices.

Although it is agreed that the educational and teaching potential of the internet and social media must be further explored, the emphasis on these positive aspects appears forced and obscures negative effects of hyperconnection, now widely demonstrated in the literature, particularly from 2020 onwards .

The main and important gap that emerges is the lack of subdivision and analysis of the 89 studies on the basis of their scientific methodology. Furthermore, it appears risky and limiting to close the interpretation of the results of these 89 studies within a predefined theoretical framework, in particular with respect to the observed phenomenon, which takes on different characteristics as multiple social characteristics vary, correlated to the different social, economic and demographic aspects.

The study therefore lacks an interdisciplinary vision.

The introduction must be shortened but also integrated in relation to the relationship between the social context and the use of social media with specific reference to the times and methods of their use, distress, happiness, satisfaction, quantity and quality of horizontal relationships, the relaction between of the growth of virtual relationships on real relationships, the perception of aesthetics and bodies, empathy, prosociality, self-esteem and primary, positive and negative emotions. These are the main variables that correlate with the use of social media, or their "abuse", and which produce eating disorders, relationship disorders, individualism, hedonism, nihilism, narcissism, inattention, social isolation, suicidal ideation, etc.

The introduction also requires an analysis of the influence of the family context, of the distinctions by gender (strongly important in this phenomenon) and based on the kynd of primary socialization experienced. Furthemore, an analysis of trends before and after the pandemic.

The article presents itself as a review of the research results, but lacks an overall analysis of the same. The results of the 82 articles should be divided by type of research. Statistically representative sample studies are one thing (the type of sample must also be distinguished), qualitative studies are another. In most cases it is the method that makes the difference. So it's not just about the different types of variables used or their limitedness. The contradictions found in the scientific findings on the analyzed phenomenon strongly depend on the robustness of the analyzed data, on the correct choice of method, on the methodological rigor, on the interpretative perspective (disciplinary or interdisciplinary).

The results of the article should therefore be reformulated by dividing them between: results of qualitative investigations and results of quantitative investigations; pre and post Covid-19 surveys. Furthermore, to avoid generalizations that are not very useful for identifying the strengths and weaknesses of the use of social media, it is necessary to specify the socio-demographic characteristics of the subjects to whom they refer. The fundamental variables of analysis are gender, age, breadth and quality of relationships in the peer network, time spent on social media, video games and video game chats, general state of well-being (distress, emotions, self-esteem).

The absence of this information constitutes the limitation of this study.

I really thank the authors very much for their attention and study on this important problem.

Author Response

Response to Reviewer 2 Comments

Summary

Thank you for taking the time to review this manuscript and for your detailed feedback. Please find the responses below and the corresponding revisions highlighted in the re-submitted file.

Comment 1: The main and important gap that emerges is the lack of subdivision and analysis of the studies on the basis of their scientific methodology.

Response 1: We agree with this comment and have now included information for each of the studies that outlines the scientific methodology (in the Appendix starting from page 16)

Comment 2: It appears risky and limiting to close the interpretation of the results of these 89 studies within a predefined theoretical framework, in particular with respect to the observed phenomenon, which takes on different characteristics as multiple social characteristics vary, correlated to the different social, economic and demographic aspects.

Response 2: Thank you for highlighting this concern. We agree that adolescent social media use is a complex phenomenon encompassing many different characteristics correlated to different social, economic, and demographic aspects. Scholars recognise this too and note that the complexity associated with social media use causes confusion when endeavouring to make sense of the current literature (e.g., Davis et al., 2020; Granic et al., 2020; Hamilton et al., 2022; Weinstein & James, 2022). To address this concern digital wellbeing scholars (e.g., Natasha Parent and Kross & colleagues) recommend applying a theoretical approach to make sense of the disparate, divergent and often contradictory findings. Thus, in line with this recommendation we have chosen to apply a SDT framework to examine adolescent social media use. We have provided a rationale for applying SDT as a framework in the original submission (page 1 lines 9-12, page 3 lines 118-134, page 5 lines 208-215). However on reflection of the reviewer’s feedback, we felt it may be useful to further clarify the importance of adopting a theoretical framework. As such, we have now included the following information in the manuscript (page 3 lines 99-101):

 “Ryan and Deci [43] note that facts without theoretical extension have little prescriptive value. Hence, this review applies a SDT framework with the aim of translating findings into practical and actionable agendas”

Comment 3: The introduction must be shortened but also integrated in relation to the relationship between the social context and the use of social media with specific reference to the times and methods of their use, distress, happiness, satisfaction, quantity and quality of horizontal relationships, the reaction between of the growth of virtual relationships on real relationships, the perception of aesthetics and bodies, empathy, prosociality, self-esteem and primary, positive and negative emotions. These are the main variables that correlate with the use of social media, or their "abuse", and which produce eating disorders, relationship disorders, individualism, hedonism, nihilism, narcissism, inattention, social isolation, suicidal ideation, etc. The introduction also requires an analysis of the influence of the family context, of the distinctions by gender (strongly important in this phenomenon) and based on the kynd of primary socialization experienced. Furthemore, an analysis of trends before and after the pandemic.

Response 3: We agree that these variables/factors are important issues that definitely warrant further research attention. However, we believe they are beyond the scope of this review which is to analyse current literature through a SDT framework. The factors the reviewer highlights are extremely relevant with regards to adolescent social media use and deserve thorough investigation that falls beyond the objective of the current review.

Comment 4: The article presents itself as a review of the research results, but lacks an overall analysis of the same. The results of the 82 articles should be divided by type of research. Statistically representative sample studies are one thing (the type of sample must also be distinguished), qualitative studies are another. In most cases it is the method that makes the difference. So it's not just about the different types of variables used or their limitedness. The contradictions found in the scientific findings on the analyzed phenomenon strongly depend on the robustness of the analyzed data, on the correct choice of method, on the methodological rigor, on the interpretative perspective (disciplinary or interdisciplinary).

Response 4: Thank you for highlighting this. We agree and have now included methodological details including study design and analyses for all the included studies within this review (please refer to the Appendix that starts on page 16).

Comment 5: Furthermore, to avoid generalizations that are not very useful for identifying the strengths and weaknesses of the use of social media, it is necessary to specify the socio-demographic characteristics of the subjects to whom they refer. The fundamental variables of analysis are gender, age, breadth and quality of relationships in the peer network, time spent on social media, video games and video game chats, general state of well-being (distress, emotions, self-esteem).

Response 5: We agree that further information for each study would be useful and have now incorporated information pertaining to gender, age, the country of origin where the study was conducted, and the type of social media or social media activity that was addressed in each study (please refer to the Appendix that starts on page 16). Although we recognise that the breadth and quality of relationships in peer networks and time spent using social media and the general state of wellbeing are important factors. Much of this information was not available within the included studies and we feel that this information is not essential to the current review questions and objectives.   

Round 2

Reviewer 2 Report

Comments and Suggestions for Authors

Dear Authors,

I appreciate the review you have conducted, but I do not consider it to be completely exhaustive. The omission of important variables, as previously indicated, results in partial scientific findings that could be disproven at any time. This is due to methodological limitations, the need for sample surveys, and the necessity of longitudinal studies. Therefore, following the previously indicated suggestions, I would ask you to re-examine the available data and possibly refer to further studies to reach conclusions that take into consideration the complexity of the phenomenon.

Thank you 

Author Response

Reviewer 2 Feedback 

Thank you for revising the manuscript and providing further feedback. 

Comment: 

The omission of important variables, as previously indicated, results in partial scientific findings that could be disproven at any time. This is due to methodological limitations, the need for sample surveys, and the necessity of longitudinal studies. Therefore, following the previously indicated suggestions, I would ask you to re-examine the available data and possibly refer to further studies to reach conclusions that take into consideration the complexity of the phenomenon.

Response: In line your feedback we have addressed the limitations you noted. In addition to outlining the methodological approaches (cross-sectional, survey, longitudinal etc) of each study in the Appendix, we have now highlighted In the limitations section (lines 551-555 and 558-561) that many of the studies included in the review were cross-sectional which needs to be considered when interpreting findings. We also emphasised the notable lack of longitudinal studies which highlights a gap in the literature that needs to be addressed within future research and how further research is needed to address social media use in regards to clinical populations and/or problematic use.
